# Geographical and Meteorological Evaluations of COVID-19 Spread in Iran

**Gholamreza Goudarzi** [1,2], **Ali Akbar Babaei** [1], **Mohammad Javad Mohammadi** [1], **Vafa Hamid** [3] **and Heydar Maleki** [1,3,*]

[1] Air Pollution and Respiratory Diseases Research Center, Ahvaz Jundishapur University of Medical Sciences, Ahvaz 61357-15794, Iran; ghgoodarzi@ajums.ac.ir (G.G.); babaei-a@ajums.ac.ir or ababaei52@gmail.com (A.A.B.); javad.sam200@gmail.com (M.J.M.)

[2] Environmental Technologies Research Center (ETRC), Ahvaz Jundishapur University of Medical Sciences, Ahvaz 61357-15794, Iran

[3] Department of Environmental Health Engineering, School of Health, Student Research Committee, Ahvaz Jundishapur University of Medical Sciences, Ahvaz 61357-15794, Iran; hamid.4300@yahoo.com

* Correspondence: heydarmaleki72@gmail.com or maleki.h@ajums.ac.ir

**Abstract:** Since late 2019 many people all over the world have become infected and have died due to coronavirus. There have been many general studies about the spread of the virus. In this study, new and accumulated confirmed cases (NCC and ACC), new and accumulated recovered cases (NRC and ARC), and new and accumulated deaths (ND and AD) were evaluated by geographical properties, meteorological parameters and air particulate matters between 3 April 2020 and 11 June 2020 within 15 provinces in Iran. Meteorological parameters, air particulate matters and COVID-19 data were collected from Iran Meteorological Organization, the Environmental Protection Agency and Aftabnews website, respectively. The results of the study show that provinces in dry lands (i.e., Kerman and South Khorasan) not only had low admission of NCC, ACC, ARC and AD but also presented lower rates of NCC, ACC and AD per $10^5$ population. Air temperature showed positive and significant correlation with the number of COVID-19 cases. This is because of hot outdoor air especially in costal and equatorial regions that forces people to stay in closed environments with no ventilation and with closed-cycle air conditioners. Maximum air pressure was found to be the most frequent (66%) and significant parameter correlating with health outcomes associated with COVID-19. The most engaged province in this study was Khuzestan, while provinces in dry lands (i.e., Kerman and South Khorasan) showed low number of health endpoints associated with COVID-19. The highest rate of accumulated and new recovered cases per $10^5$ population were also found in Khuzestan and Kerman provinces. North Khorasan also showed the worst rate of N&ARC/$10^5$ population. Therefore, air temperature, dry lands and population were the most important factors for the control of coronavirus spread.

**Keywords:** geographical properties; meteorological parameters; number and rates of COVID-19 cases; Iran

## 1. Introduction

In late 2019 a novel Severe Acute Respiratory Syndrome coronavirus emerged in Wuhan, China which was called COVID-19 by World Health Organization. At first, on 30 January 2020, the new health endpoint was considered to be a public health emergency then on 11 February 2020, COVID-19 was officially stated to be a pandemic [1]. The rapid transmission, wide range, strong infectivity and the long incubation period ranging between 2 and 14 days are the most important factors increasing the severity of COVID-19 outcomes [2–4]. The common symptoms of the disease are fever, dry cough and shortness of breath [5]. Good physical condition is a considerable parameter for prevention of getting infected by the virus so that older people and people with existing health endpoints

including hypertension, chronic obstructive pulmonary disease, cerebrovascular disease and diabetes are the most-at-risk portion of the population [6,7].

Air pollution is one of the factors through which the spread of the virus is facilitated [8–10]. Setti et al. have shown that the boost of COVID-19 spread in northern parts of Italy is correlated with particulate matter [11]. They also believe that high level of relative humidity increased the diffusion rate of COVID-19, though high level of air temperature decreased it. Supporting results were found in a study in Turkey for air temperature in which low levels of air temperature caused higher COVID-19 cases over the course of the same day [12]. However, in that study, higher levels of humidity resulted in lower number of COVID-19 cases. $PM_{2.5}$, $PM_{10}$, CO, $NO_2$ and $O_3$ are known as criteria air pollutants that have shown direct associations with new confirmed COVID-19 cases in China [13]. A similar study in Italy illustrated that $PM_{2.5}$ could increase the rate of COVID-19 cases [14]. Another study in India has shown that a 10 $\mu g/m^3$ increase in fine atmospheric particulate matters ($PM_{2.5}$) with a lag of 0 to 14 days is associated with higher number of new COVID-19 cases [15]. Wu et al., demonstrated in 2020 through observation of data from 3000 counties in the US that if the concentration of $PM_{2.5}$ increases by 1 $\mu g/m^3$ the death rate of COVID-19 increases by 8% [16]. Indoor fine and coarse particulate matter in Middle Eastern countries has facilitated the transmission of the virus [17], while lower levels of criteria air pollutants and relative humidity, and higher air temperature decreased the spread of COVID-19 in Kuala Lumpur, Malaysia [18].

In Iran, it has been shown that $PM_{2.5}$ reduced around 40% during quarantine period in Tehran [19]. An opposite study in Tehran from 21 March to 21 April 2020 showed that during lockdown $PM_{2.5}$ and $O_3$ increased and levels of CO, $NO_2$, $SO_2$ and $PM_{10}$ decreased [20]. Another study in Tehran, Iran showed that fine and coarse particulate matter did not decrease during the COVID-19 outbreak (20 February to 2 April 2020) [21]. Asna-ashary et al., in 2020 unexpectedly reported that a reduction of air pollution in Iran could cause increasing number of COVID-19 cases [22]. Positive and significant correlation was found between air pollutants (i.e., $PM_{2.5}$ and $NO_2$) and COVID-19 cases in Iran [23]. Ahmadi et al., 2020 showed that air temperature and rainfall do not have any significant relationship with the fluctuations of the COVID-19 cases [24]. While lower values of relative humidity, wind speed and solar radiation were attributed to higher rates of virus exposure. The same results were arrived at by Sahin, and Setti et al. and were also arrived at regarding air temperature in Iran by Jahangiri et al. [11,12,25]. Since these studies evaluating the interactions of COVID-19 cases and atmospheric parameters are general and limited to Iran in particular, this study aims to evaluate the possible relationship between visibility and meteorological parameters with different types of COVID-19 cases. These include new confirmed cases (NCC), accumulated confirmed cases (ACC), new recovered cases (NRC), accumulated recovered cases (ARC), new deaths (ND) and accumulated deaths (AD). The current study was conducted in 15 provinces over the course of three months.

## 2. Methodology

### 2.1. Study Area

Iran is a country with an area of $1.648 \times 10^6$ $km^2$ and a population of over 80 million. It is located in the Middle East so that it connects Asia to Europe by land. The Caspian Sea to the north and the Persian Gulf and Gulf of Oman to the south have limited land transportation to Iran. Iraq, Turkey, Armenia, Azerbaijan, Turkmenistan, Afghanistan and Pakistan are the countries in vicinity to Iran (Figure 1). East and south of Iran are mainly dry lands while northern and western parts of the country are mostly mountainous. In this study, 15 provinces were analyzed in which Khuzestan, Bushehr and Hormozgan are recognized as coastal regions, Kerman and South Khorasan as dry lands and the rest of the provinces as mountainous areas. The most populated provinces in the current study are Khuzestan, Fars and West Azerbaijan (Table 1).

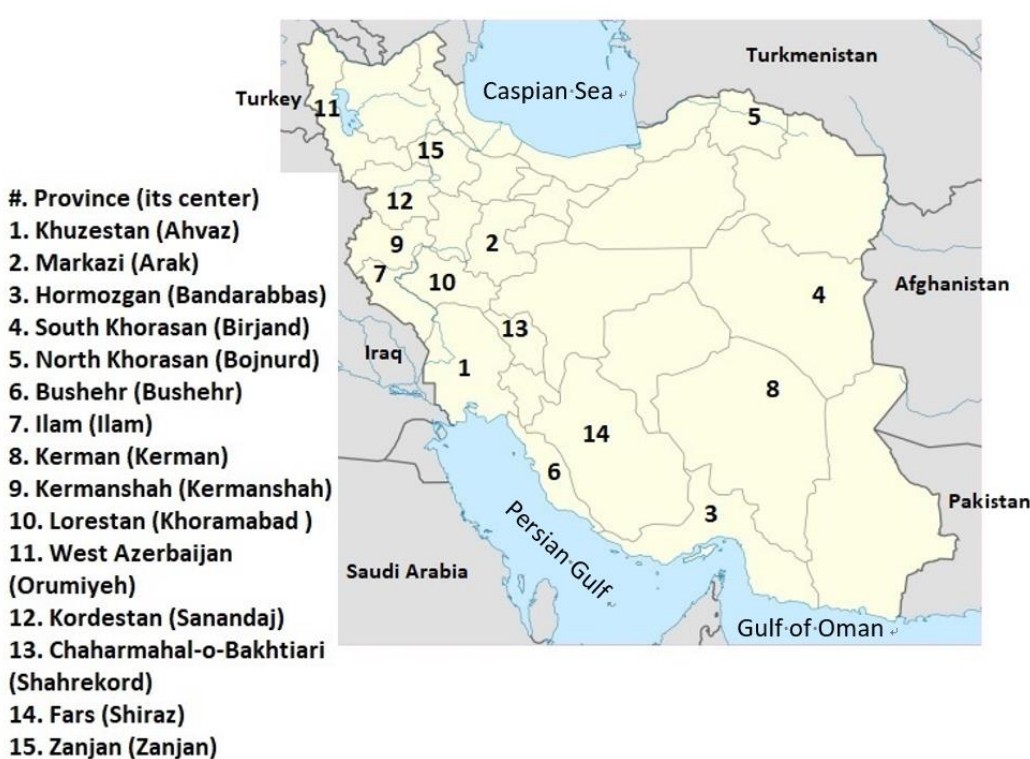

**Figure 1.** Geographical locations of Iran and studied provinces in this study.

**Table 1.** General urban and geographical information about the provinces in Iran.

| Province | Capital | Lat. (N) | Lon. (E) | Population | Geographical Feature |
|---|---|---|---|---|---|
| Khuzestan | Ahvaz | 31.3 | 48.7 | 3,554,205 | Plain & costal |
| Markazi | Arak | 34.1 | 49.7 | 1,099,764 | Mountainous |
| Hormozgan | Bandarabbas | 27.2 | 56.3 | 971,822 | Costal |
| South Khorasan | Birjand | 32.9 | 59.2 | 453,827 | Dry land |
| North Khorasan | Bojnurd | 37.5 | 57.3 | 484,346 | Mountainous |
| Bushehr | Bushehr | 29.0 | 50.8 | 835,955 | Costal |
| Ilam | Ilam | 33.6 | 46.4 | 395,263 | Mountainous |
| Kerman | Kerman | 30.3 | 57.1 | 1,858,587 | Dry land |
| Kermanshah | Kermanshah | 34.3 | 47.1 | 1,468,615 | Mountainous |
| Lorestan | Khorramabad | 33.5 | 48.3 | 1,134,908 | Mountainous |
| West Azerbaijan | Orumiyeh | 37.6 | 45.1 | 2,136,203 | Mountainous |
| Kordestan | Sanandaj | 35.3 | 47.0 | 1,134,229 | Mountainous |
| Chaharmahal-o-Bakhtiari | Shahrekord | 32.3 | 50.9 | 607,444 | Mountainous |
| Fars | Shiraz | 29.6 | 52.5 | 3,401,675 | Mountainous |
| Zanjan | Zanjan | 36.7 | 48.5 | 711,177 | Mountainous |

*2.2. Data Gathering*

COVID-19 data, including new confirmed cases (NCC), accumulated confirmed cases (ACC), new recovered cases (NRC), accumulated recovered cases (ARC), new deaths (ND) and accumulated deaths (AD), were reported by the officials of the medical sciences universities and collected from https://aftabnews.ir (accessed on 3 April 2020). The data were gathered from 3 April to 11 June 2020 In Iran. COVID-19 information was not reported for other provinces which are missing in this study. Visibility (m) and meteorological parameters including air pressure (hPa), air temperature (°C), relative humidity (%), dew point (°C), wind speed (m/s) and wind direction (°) were collected from the Iran Meteorological Organization https://www.irimo.ir(accessed on 3 April 2020) over the same timeframe as for the COVID-19 data. In this study, meteorological parameters of the capital cities of the provinces were considered for the whole province. Meteorological variables

were recorded in synoptic stations with 3-h time step. These data were reported based on coordinated universal time (UTC = Local time − 3.5 h). Finally, air particulate matter ($PM_{2.5}$ and $PM_{10}$) data were collected for Ahvaz city through the air pollution monitoring system http://aqms.doe.ir/(accessed on 3 April 2020). For more accurate analysis, mean, minimum and maximum of visibility and meteorological parameters were evaluated with NCC, ACC, NRC, ARC, ND and AD.

*2.3. Statistical Analysis*

Pearson correlation evaluates the significance of the relationship between two quantitative parameters using Equation (1):

$$r = \frac{n(\sum xy) - (\sum x)(\sum y)}{\sqrt{(n(\sum x^2) - (\sum x)^2)(n(\sum y^2) - (\sum y)^2)}} \tag{1}$$

where x and y are variables and n is number of variables. In the current study, Pearson correlation was conducted between different types of COVID-19 cases (i.e., NCC, ACC, NRC, ARC, ND and AD) and meteorological parameters as well as particulate matters. Mean, minimum and maximum for air pressure (P), air temperature (T), dew point temperature (Td), relative humidity (RH), visibility (V), wind speed (WS), and wind direction (WD) with a lag of 0–14 days were correlated with NCC, ACC, NRC, ARC, ND and AD in order to find the most parameters that most correlated with direct or reverse relationships. The other aim was to find out which parameter showed the most significant correlation with the shortest lag day.

## 3. Results and Discussion

By analyzing Tables 2–5, we can find that following content. NCC, ACC and ARC in Khuzestan province were lower than Fars and West Azerbaijan during April 2020. However, they increased with higher rates of about 8.6, 621.2 and 580.4 cases per $10^5$ population (Table 5) in Khuzestan during May, respectively. AD and ND profiles in Khuzestan were only lower than Markazi and West Azerbaijan at the beginning of April 2020, respectively (Figures 2 and 3). Among the fifteen provinces, Khuzestan province showed the highest infected cases with a significant difference. Through the three types of COVID-19 cases, provinces in dry lands (i.e., Kerman and South Khorasan) showed the lowest infectious people, whereas other provinces in coastal and mountainous regions presented the majority of COVID-19 data in this study.

**Table 2.** The correlation coefficient (r) between NCC and mean, minimum (min) and maximum (max) of particulate matter, visibility and meteorological parameters with the 14-lag day in Iran. #D refers to the number of delayed days.

| Provinces | PM$_{10}$min & 9D | PM$_{10}$mean & 7D | PM$_{10}$ & #D | PM$_{2.5}$min & 3D | PM$_{2.5}$mean & 8D | PM$_{2.5}$max & 7D | |
|---|---|---|---|---|---|---|---|
| Ahvaz [C] | 0.565 ** | 0.308 * | - | 0.443 ** | 0.542 ** | 0.348 ** | |
| | Pmax & 0D | Tmean & 4D | RHmean & 2D | Vmin & 8D | Tdmin & 1D | WDmean & 2D | WSmin & 1D |
| Ahvaz [C] | −0.817 ** | 0.910 ** | −0.806 ** | −0.334 ** | −0.423 ** | 0.541 ** | 0.259 * |
| | Pmin & 2D | Tmean & 14D | RHmax & 11D | Vmean & 0D | Tdmax & 0D | WDmax & 2D | WSmean & 0D |
| Bandarabbas [C] | −0.847 ** | 0.772 ** | −0.614 ** | −0.458 ** | 0.828 ** | −0.374 ** | 0.412 ** |
| | Pmax & 3D | Tmin & 1D | RHmax & 12D | Vmax & 7D | Tdmean & 4D | WDmin & 3D | WSmin & 4D |
| Bushehr [C] | −0.637 ** | 0.597 ** | 0.292 * | −0.508 ** | 0.663 ** | 0.373 ** | 0.372 ** |
| | Pmax & 9D | Tmin & 9D | RHmax & 9D | Vmin & 4D | Tdmax & 6D | WD & #D | WS & #D |
| Arak [M] | −0.459 ** | 0.519 ** | −0.498 ** | 0.358 * | 0.454 ** | - | - |
| | Pmean & 10D | Tmin & 14D | RHmin & 14D | Vmin & 9D | Tdmax & 7D | WD & #D | WS & #D |
| Bojnurd [M] | −0.551 ** | 0.594 ** | −0.432 ** | 0.377 ** | 0.599 ** | - | - |
| | Pmax & 2D | Tmean & 3D | RHmax & 4D | V & #D | Tdmean & 5D | WDmean & 6D | WSmean & 1D |
| Ilam [M] | −0.632 ** | 0.679 ** | −0.650 ** | - | −0.416 ** | 0.346 ** | 0.307 ** |
| | Pmax & 3D | Tmax & 7D | RHmax & 2D | Vmin & 9D | Tdmean & 0D | WD & #D | WS & #D |
| Kermanshah [M] | −0.615 ** | 0.927 ** | −0.890 ** | 0.303 * | −0.774 ** | - | - |
| | Pmax & 1D | Tmean & 14D | RHmean & 13D | Vmin & 0D | Tdmax & 0D | WDmax & 7D | WS & #D |
| Khorramabad [M] | −0.637 ** | 0.707 ** | −0.687 ** | 0.521 ** | −0.571 ** | −0.373 ** | - |
| | Pmax & 6D | Tmax & 6D | RHmean & 6D | Vmin & 8D | Tdmax & 13D | WD & #D | WSmin & 5D |
| Orumiyeh [M] | −0.362 ** | 0.757 ** | −0.633 ** | 0.447 ** | 0.650 ** | - | −0.274 * |
| | Pmax & 10D | Tmean & 14D | RHmax & 9D | V & #D | Tdmean & 4D | WD & #D | WS & #D |
| Sanandaj [M] | −0.550 ** | 0.738 ** | −0.779 ** | - | −0.434 ** | - | - |
| | Pmax & 5D | Tmean & 6D | RHmean & 5D | V & #D | Tdmean & 4D | WDmax & 1D | WSmean & 14D |
| Shahrekord [M] | −0.420 ** | 0.545 ** | −0.461 ** | - | −0.255 * | 0.363 ** | −0.299 * |
| | P & #D | T & #D | RH & #D | Vmax & 9D | Tdmin & 2D | WDmax & 7D | WS & #D |
| Shiraz [M] | - | - | - | 0.255 * | −0.273 * | 0.291 * | - |
| | Pmax & 0D | Tmean & 0D | RHmean & 0D | Vmin & 6D | Tdmin & 14D | WDmax & 13D | WS & #D |
| Zanjan [M] | 0.624 ** | −0.750 ** | 0.598 ** | −0.420 ** | −0.574 ** | −0.321 ** | - |
| | Pmin & 0D | Tmin & 0D | RHmax & 6D | Vmean & 12D | Tdmax & 7D | WDmean & 10D | WSmean & 1D |
| Birjand [D] | −0.343 ** | 0.362 ** | −0.259 * | 0.452 ** | −0.362 ** | −0.370 ** | 0.429 ** |
| | Pmin & 2D | Tmax & 4D | RHmax & 3D | Vmean & 10D | Tdmax & 5D | WDmean & 0D | WSmin & 10D |
| Kerman [D] | −0.682 ** | 0.635 ** | −0.504 ** | 0.668 ** | −0.349 ** | −0.345 ** | −0.267 * |

* Correlation is significant at the 0.05 level (2-tailed). ** Correlation is significant at the 0.01 level (2-tailed). [C]: Coastal; [M]: Mountainous; [D]: Dry lands.

**Table 3.** The correlation coefficient (r) between NRC and mean, minimum (min) and maximum (max) of particulate matter, visibility and meteorological parameters with the 14-lag day in Iran. #D refers to the number of delayed days.

| Provinces | $PM_{10}min$ & 11D | $PM_{10}mean$ & 8D | $PM_{10}max$ & 10D | $PM_{2.5}min$ & 6D | $PM_{2.5}mean$ & 10D | $PM_{2.5}max$ & 10D | - |
|---|---|---|---|---|---|---|---|
| Ahvaz [C] | 0.554 ** | 0.565 ** | 0.483 ** | 0.480 ** | 0.617 ** | 0.570 ** | - |
| Ahvaz [C] | Pmax & 2D −0.835 ** | Tmean & 5D 0.910 ** | RHmean & 2D −0.813 ** | Vmin & 13D −0.309 * | Tdmin & 1D −0.461 ** | WDmean & 2D 0.587 ** | WS & #D - |
| Bandarabbas [C] | P & #D - | T & #D - | RHmean & 0D −0.618 ** | Vmax & 13D 0.625 ** | Tdmin & 0D −0.606 ** | WDmax & 8D −0.622 ** | WSmax & 12D −0.481 * |
| Bushehr [C] | Pmax & 4D −0.569 ** | Tmin & 6D 0.543 ** | RHmax & 5D −0.259 * | Vmax & 8D −0.599 ** | Tdmean & 9D 0.623 ** | WDmin & 6D 0.445 ** | WSmin & 5D 0.322 * |
| Arak [M] | Pmax & 0D 0.739 ** | Tmean & 2D −0.602 * | RHmax & 11D −0.518 * | V & #D - | Tdmax & 1D −0.566 * | WDmin & 13D 0.589 * | WS & #D - |
| Bojnurd [M] | Pmin & 12D −0.321 * | Tmin & 14D 0.355 * | RH & #D - | Vmean & 4D 0.571 ** | Tdmin & 14D 0.399 ** | WDmin & 7D 0.357 * | WSmin & 8D 0.376 ** |
| Khorramabad [M] | Pmax & 0D −0.633 ** | Tmax & 14D 0.741 ** | RHmean & 14D −0.708 ** | Vmin & 0D 0.523 ** | Tdmean & 4D −0.712 ** | WDmax & 0D 0.395 * | WSmax & 4D 0.389 * |
| Orumiyeh [M] | Pmax & 13D −0.456 ** | Tmin & 7D 0.394 ** | RHmean & 13D −0.489 ** | Vmin & 5D 0.380 * | Tdmax & 6D 0.398 ** | WDmin & 5D −0.448 ** | WSmean & 2D −0.307 * |
| Sanandaj [M] | Pmean & 4D −0.410 ** | Tmean & 8D 0.689 ** | RHmax & 7D −0.635 ** | V & #D - | Tdmean & 1D −0.488 ** | WDmin & 1D −0.271 * | WS & #D - |
| Shahrekord [M] | Pmax & 5D −0.411 ** | Tmax & 6D 0.493 ** | RHmean & 3D −0.381 ** | Vmin & 13D 0.284 * | Td & #D - | WDmax & 1D 0.339 ** | WS & #D - |
| Shiraz [M] | P & #D - | T & #D - | RHmin & 2D 0.597 ** | Vmean & 2D −0.357 ** | Tdmax & 5D −0.263 * | WDmin & 5D 0.576 ** | WSmin & 5D 0.686 ** |
| Zanjan [M] | Pmax & 0D 0.453 ** | Tmin & 3D −0.537 ** | RHmin & 9D 0.412 ** | Vmin & 1D −0.355 ** | Tdmax & 8D −0.424 ** | WDmax & 11D −0.309 * | WS & #D - |
| Birjand [D] | Pmin & 0D −0.357 ** | Tmin & 0D 0.360 ** | RHmin & 14D 0.251 * | Vmax & 12D 0.476 ** | Td & #D - | WDmean & 10D −0.434 ** | WSmean & 1D 0.382 ** |
| Kerman [D] | Pmean & 8D 0.452 ** | Tmax & 8D −0.526 ** | RHmin & 8D 0.379 * | Vmax & 14D 0.449 ** | Tdmax & 5D −0.506 ** | WDmean & 1D 0.456 ** | WSmin & 7D 0.382 * |

* Correlation is significant at the 0.05 level (2-tailed). ** Correlation is significant at the 0.01 level (2-tailed). [C]: Coastal; [M]: Mountainous; [D]: Dry lands.

**Table 4.** The correlation coefficient (r) between ND and mean, minimum (min) and maximum (max) of particulate matters, visibility and meteorological parameters with the 14-lag day in Iran. #D refers to the number of delayed days.

| Provinces | PM$_{10}$min & 12D | PM$_{10}$mean & 10D | PM$_{10}$ & #D | PM$_{2.5}$min & 4D | PM$_{2.5}$mean & 12D | PM$_{2.5}$max & 10D | |
|---|---|---|---|---|---|---|---|
| Ahvaz [C] | 0.532 ** | 0.336 ** | - | 0.417 ** | 0.533 ** | 0.388 ** | |
| | Pmax & 4D | Tmean & 0D | RHmean & 2D | Vmin & 10D | Tdmin & 2D | WDmean & 3D | WS & #D |
| Ahvaz [C] | −0.729 ** | 0.868 ** | −0.791 ** | −0.304 * | −0.404 ** | 0.553 ** | - |
| | Pmin & 7D | Tmax & 6D | RHmax & 13D | Vmean & 11D | Tdmax & 4D | WDmax & 10D | WSmean & 3D |
| Bandarabbas [C] | −0.613 ** | 0.503 ** | −0.356 ** | −0.361 ** | 0.603 ** | −0.553 ** | 0.380 ** |
| | Pmax & 5D | Tmin & 7D | RHmax & 5D | Vmax & 9D | Tdmean & 10D | WDmin & 7D | WSmin & 6D |
| Bushehr [C] | −0.498 ** | 0.476 ** | −0.258 * | −0.642 ** | 0.486 ** | 0.554 ** | 0.449 ** |
| | Pmax & 1D | Tmax & 9D | RHmin & 9D | Vmax & 1D | Tdmin & 9D | WDmin & 2D | WSmin & 2D |
| Arak [M] | 0.531 ** | −0.559 ** | 0.563 ** | −0.348 * | 0.397 * | 0.467 ** | 0.516 ** |
| | P & #D | T & #D | RH & #D | Vmax & 2D | Tdmin & 12D | WDmax & 1D | WS & #D |
| Bojnurd [M] | - | - | - | 0.308 * | 0.312 * | −0.323 * | - |
| | Pmean & 8D | T & #D | RHmin & 2D | Vmin & 3D | Tdmean & 2D | WD & #D | WS & #D |
| Ilam [M] | 0.275 * | - | 0.353 ** | −0.255 * | 0.327 ** | - | - |
| | Pmean & 1D | Tmean & 1D | RHmax & 6D | Vmin & 5D | Tdmin & 6D | WDmax & 6D | WSmin & 4D |
| Kermanshah [M] | −0.344 ** | 0.486 ** | −0.520 ** | 0.315 ** | −0.560 ** | 0.266 * | −0.252 * |
| | Pmax & 4D | Tmean & 4D | RHmean & 4D | Vmin & 9D | Tdmin & 12D | WDmean & 12D | WSmean & 8D |
| Khorramabad [M] | 0.383 ** | −0.343 ** | 0.333 * | −0.303 * | 0.318 * | 0.289 * | 0.311 * |
| | Pmean & 5D | T & #D | RH & #D | Vmax & 5D | Td & #D | WDmax & 13D | WSmax & 12D |
| Orumiyeh [M] | 0.390 ** | - | - | 0.285 * | - | 0.357 ** | 0.317 * |
| | P & #D | T & #D | RH & #D | Vmin & 3D | Tdmin & 4D | WDmin & 5D | WSmin & 0D |
| Sanandaj [M] | - | - | - | −0.265 * | −0.356 ** | 0.316 ** | 0.302 * |
| | Pmax & 14D | Tmax & 0D | RHmin & 0D | V & #D | Tdmin & 9D | WDmean & 5D | WSmean & 14D |
| Shahrekord [M] | 0.276 * | −0.282 * | 0.367 ** | - | −0.291 * | 0.291 * | −0.236 * |
| | Pmax & 7D | Tmin & 3D | RHmin & 6D | Vmax & 4D | Tdmean & 5D | WDmax & 1D | WSmax & 9D |
| Shiraz [M] | 0.447 ** | −0.394 ** | 0.461 ** | −0.326 ** | 0.377 ** | −0.251 * | 0.278 * |
| | Pmin & 12D | Tmin & 12D | RHmean & 2D | Vmin & 3D | Tdmin & 12D | WDmax & 11D | WSmin & 2D |
| Zanjan [M] | 0.346 ** | −0.403 ** | 0.349 ** | −0.417 ** | −0.454 ** | −0.301 * | 0.305 * |
| | Pmax & 0D | Tmax & 14D | RHmin & 0D | Vmin & 9D | Tdmin & 0D | WDmin & 7D | WSmean & 9D |
| Birjand [D] | 0.329 ** | −0.368 ** | 0.461 ** | −0.446 ** | 0.327 ** | 0.334 ** | 0.348 ** |
| | Pmax & 2D | Tmax & 3D | RHmax & 3D | Vmin & 10D | Tdmax & 6D | WD & #D | WSmax & 10D |
| Kerman [D] | −0.251 * | 0.320 ** | −0.380 ** | 0.407 ** | −0.373 ** | - | −0.255 * |

* Correlation is significant at the 0.05 level (2-tailed). ** Correlation is significant at the 0.01 level (2-tailed). [C]: Coastal; [M]: Mountainous; [D]: Dry lands.

**Table 5.** NCC, ACC, NRC, ARC, ND and AD rates per population attributed to COVID-19 in each province during 71 days in Iran.

| Province | Capital | NCC/ $10^5$ Population $\times$ Day | NRC/ $10^5$ Population $\times$ Day | ND/Day | ACC/ $10^5$ Population | ARC/ $10^5$ Population | AD/ $10^5$ Population |
|---|---|---|---|---|---|---|---|
| Khuzestan | Ahvaz | 8.6 | 8.3 | 9 | 621.2 | 580.4 | 20.6 |
| Bushehr | Bushehr | 3.7 | 1.5 | 0 | 266.8 | 108.1 | 4.1 |
| Hormozgan | Bandarabbas | 9.6 | 1.3 | 1 | 681.7 | 358.3 | 10.4 |
| Markazi | Arak | 2.1 | 1.4 | 1 | 227.8 | 132.8 | 16.2 |
| North Khorasan | Bojnurd | 6.7 | 1.7 | 1 | 509.6 | 171.8 | 29.1 |
| Ilam | Ilam | 3.7 | - | 1 | 338.8 | - | 18.5 |
| Kermanshah | Kermanshah | 7.4 | - | 2 | 515.7 | - | 11.5 |
| Lorestan | Khorramabad | 7.1 | 7.6 | 2 | 485.7 | 466.2 | 15.1 |
| West Azerbaijan | Orumiyeh | 5.1 | 1.6 | 3 | 384.9 | 221.0 | 12.6 |
| Kordestan | Sanandaj | 5.3 | 2.5 | 2 | 407.3 | 185.1 | 15.9 |
| Chaharmahal-o-Bakhtiari | Shahrekord | 1.8 | 1.8 | 0 | 147.8 | 137.5 | 5.3 |
| Fars | Shiraz | 2.6 | 1.5 | 1 | 209.6 | 179.0 | 3.5 |
| Zanjan | Zanjan | 1.3 | 1.2 | 1 | 176.0 | 146.4 | 19.5 |
| South Khorasan | Birjand | 1.7 | 1.7 | 0 | 188.0 | 172.3 | 10.4 |
| Kerman | Kerman | 2.1 | 0.3 | 1 | 158.7 | 74.2 | 5.1 |

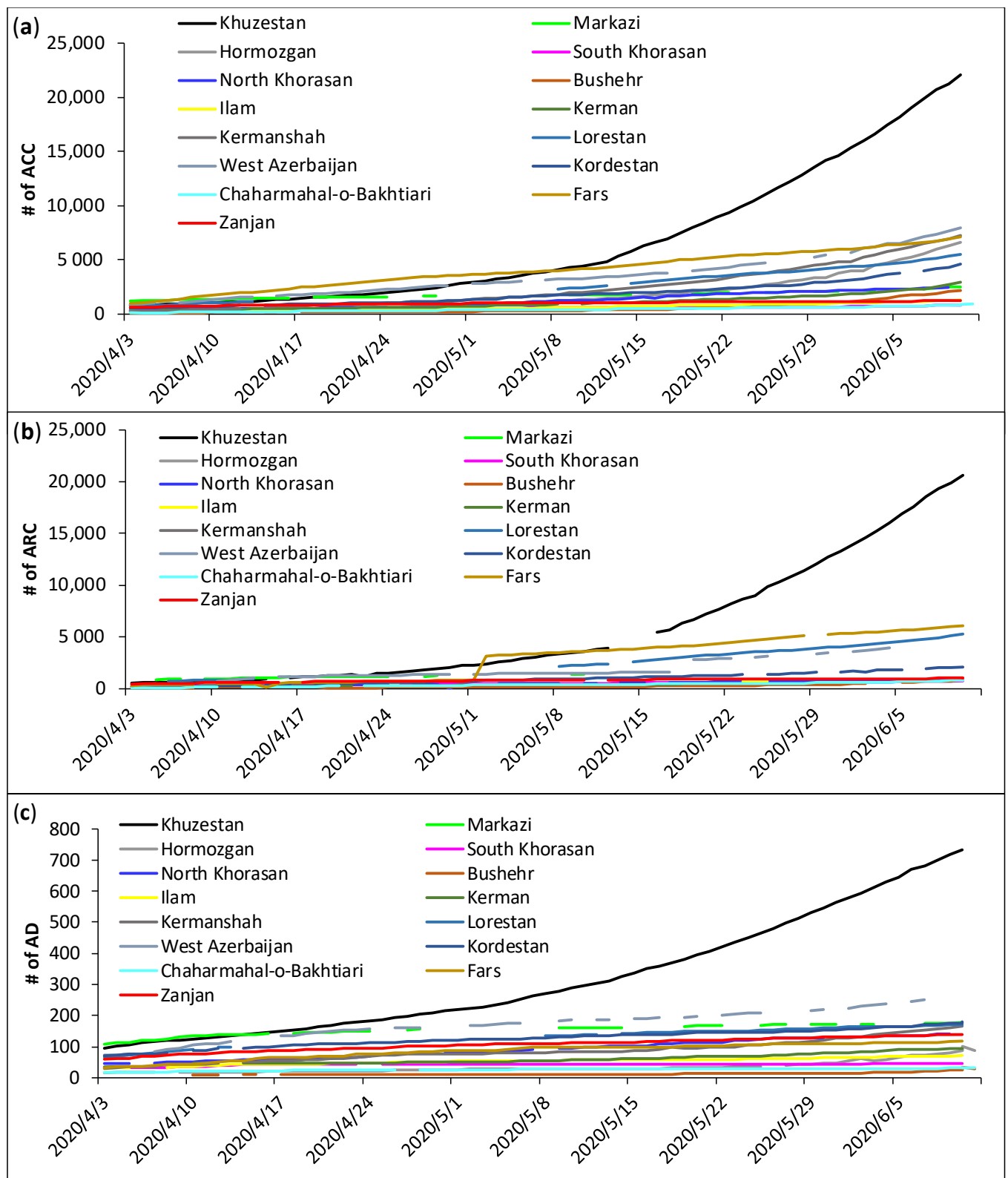

**Figure 2.** Temporal profiles of (**a**) ACC, (**b**) ARC and (**c**) AD attributed to COVID-19 from April to June 2020 in Iran.

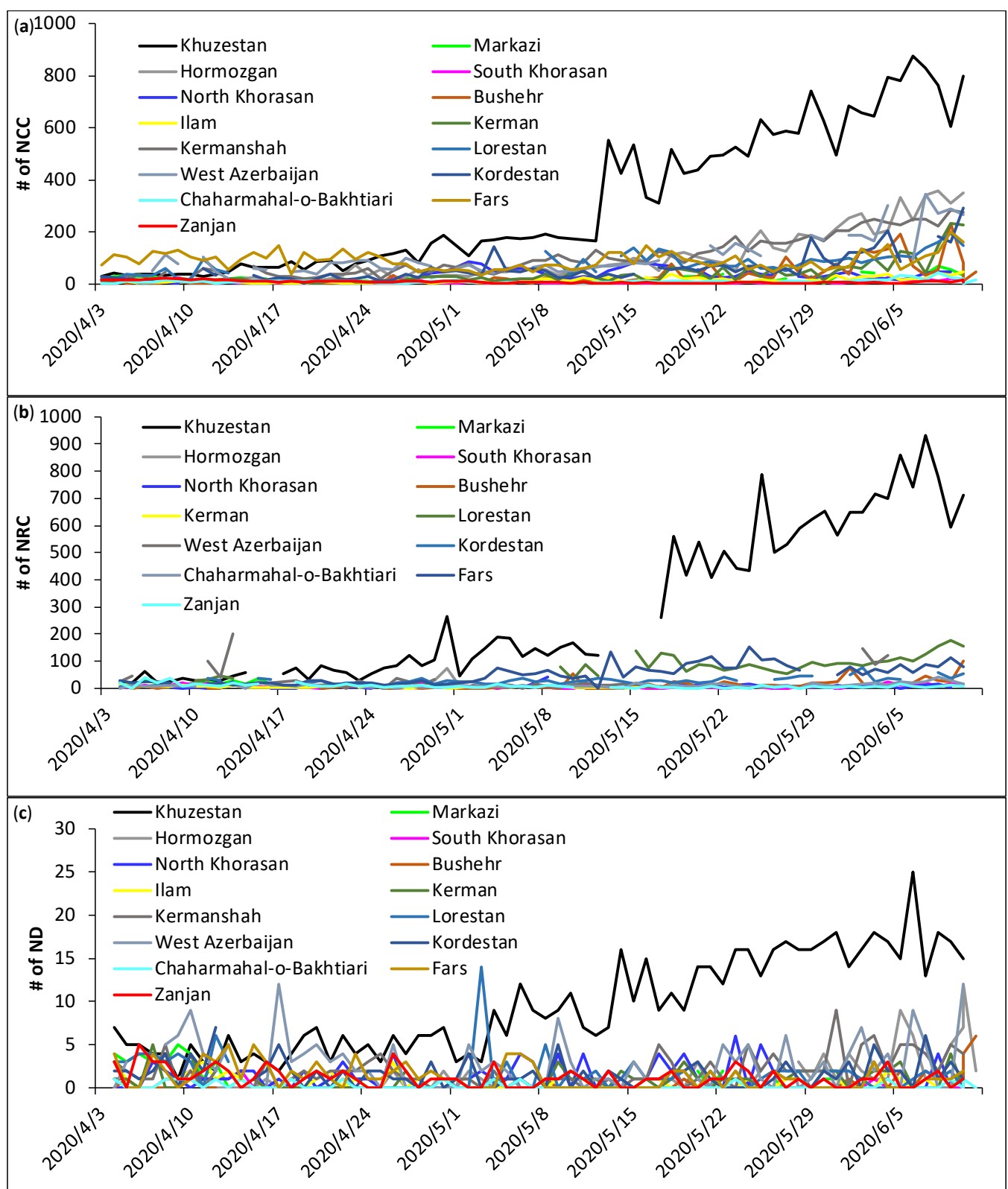

**Figure 3.** Temporal profiles of (**a**) NCC, (**b**) NRC and (**c**) ND attributed to COVID-19 from April to June 2020 in Iran.

Positive and significant correlation was found between particulate matters ($PM_{2.5}$ and $PM_{10}$) and different health endpoints associated with COVID-19 in Khuzestan province. Amato et al. also showed positive tests for the presence of SARS-CoV-2 through 57% of

total suspended particles samples [26]. In the mentioned study, a negative and significant relationship was found between relative humidity and air temperature with the prevalence of coronavirus genome. Many other studies have also shown the same result about the reverse correlation between air temperature and spread of coronavirus [11,12,18,24,25]. Similar to the study of Zoran et al., the relationship between air temperature and spread of SARS-CoV-2 in the current study was positive in Iran [27]. This event might be due to the way in which, in coastal regions, air conditioners work continuously from spring to autumn in all buildings through a closed cycle of indoor air with no ventilation [28]. If somebody is in the room that is infected by coronavirus, then it can spread very fast. Increased air temperature in the outdoor environment would not be able to prevent the spread of virus [27]. In general, significant reverse relationships were found between air pressure and relative humidity with the prevalence of coronavirus. The possible hypothesis for air pressure is that it might increase the number of viruses per cubic meters of air. Ahmadi et al. and Sahin have also shown the same results about relative humidity [12,24]. Setti et al. and Suhaimi et al. showed contrasting findings with this study about relative humidity [11,18]. Relative humidity, which is low in cities with a dry climate such as Kerman and Birjand, could be the reason for the low rate of coronavirus spread. Among all meteorological parameters in Tables 2–4, maximum air pressure and minimum visibility were the most frequent parameters showing the most significant relationships with NCC, NRC and ND. Among the 38 significant relationships between air pressure and COVID-19 cases, maximum air pressure represented 25 (66%) significant correlations. Similarly, minimum visibility showed 20 (54%) significant correlations out of 37 strong relationships of visibility and COVID-19 health endpoints. Significant correlation between fine particulate matter ($PM_{2.5}$) and different covid-19 health outcomes (NCC, NRC and ND) occurred through shorter lag day than coarse particulate matters ($PM_{10}$). Multiple regression using $PM_{2.5}$ showed a much stronger model for estimation of COVID-19 health outcomes than using $PM_{10}$ in Poland [29]. The shortest lag day for the significant spread of SARS-CoV-2 demonstrated by air pressure was about 4.5 days. The average lag day between visibility and the COVID-19 health endpoints (NCC, NRC and ND) was 6.8 days in Iran. Surprisingly, this delaying time was as the same as 10.3 days for visibility and particulate matters in order to fasten the spread of the virus in Khuzestan province. The strongest correlation was often shown between air temperature and the cases attributed to COVID-19. Air temperature trends in coastal provinces in the south of Iran were around to be 10 °C higher than other parts of the country (Figure 4). The correlation coefficients between health outcomes attributed to coronavirus, and air temperature and pressure were stronger in southern coasts of Iran than in mountainous and dry climates. Ahmadi et al. have shown that low level of wind speed can be associated with higher exposure to the virus in Iran [24]. However, in this study, wind speed showed a neutral relationship with different health endpoints associated to coronavirus probably due to the full range consideration of wind speed levels.

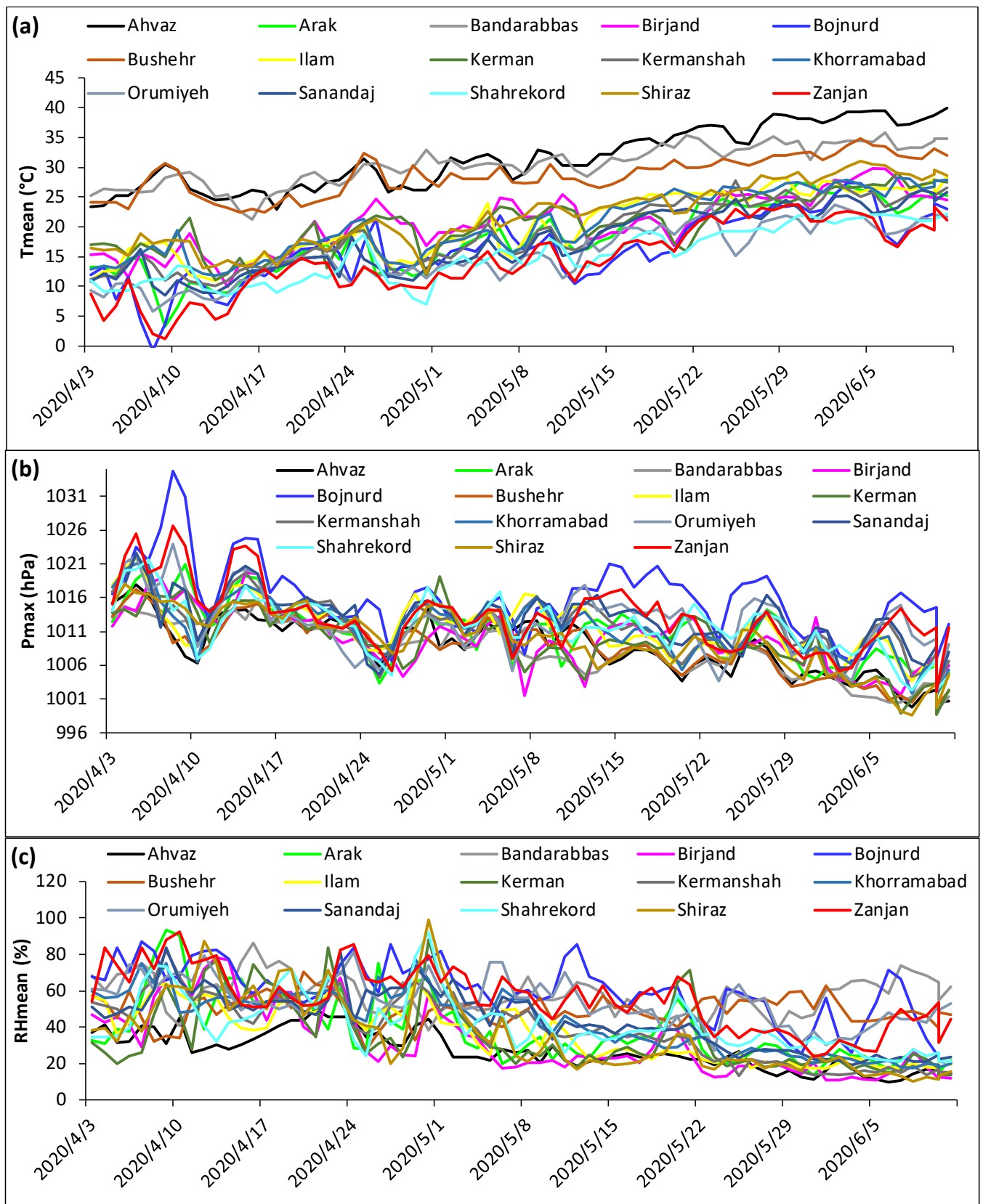

**Figure 4.** Daily variations of (**a**) mean air temperature (Tmean), (**b**) maximum air pressure (Pmax), (**c**) mean relative humidity (RHmean) with lag of 0 days in Iran during the study period.

Schools were closed in Iran from March 2020 to March 2022. At the beginning of the closure, lessons were taught to children through national television. Lack of sufficient infrastructure for virtual training and limited number cellphones, computers and so on were the main reasons for live TV teaching before the Shad application was implemented by the Ministry of Education so that students had access to educational contents. During the closure, students sometimes went to schools during exam times. In addition, each class was divided into two or three groups with different class times during the closure because some lessons where difficult to teach virtually and there were some places where students had no electronic devices to learn through virtual training. Students who became infected with coronavirus, had similar symptoms to COVID-19 or had previous health problems were not recommended to come to schools in all situations [30]. Vaccination was started from Winter 2021 by the Astrazeneca vaccine through four steps. At first, personnel who fought against COVID-19 in hospitals (doctors, nurses and so on) and elder people who had severe health problems were vaccinated from Winter 2021. The estimated amount of people in these groups was about 1.3 million. Secondly, all elderly people over 65 years old (6 million people), and adults and youths aged 16 to 64 years old with previous health problems (6 million people) were vaccinated from Spring 2021. The third group included people in crowded centers (2 million), all people who were aged between 55–64 years old (5 million people) and people who had necessary jobs (approximately 12 million people). This group was vaccinated from July 2021. Finally, the whole population were vaccinated from Winter 2022 [31].

Figure 5 was conducted to show the quantitative relationships (equations) and co-efficients of determination ($R^2$) between NCC cases and influencing parameters in Iran. The strong relationship between ($R^2 \geq 0.7$) meteorological parameters and NCC cases was found in Ahvaz through mean air temperature; in Bandarabbas through mean air temperature and minimum air pressure; and in Kermanshah through maximum air temperature and maximum relative humidity. Therefore, it seems air temperature has a big role on the infection rate of COVID-19 between people in the country especially in costal and equatorial regions. Multiple regression was conducted through $PM_{2.5}$, population density and the number of laboratory COVID-19 tests in Polish provinces in order to estimate the new number of infections. The model, including the three parameters, was reliable ($R^2 > 0.7$) only in December while it was not strong enough though other months. A similar model, including $PM_{10}$, population density and the number of laboratory COVID-19 tests in Polish provinces, could not present a strong model to estimate the new number of infections [29]. In this study, $PM_{10}$ and $PM_{2.5}$ showed significant and positive relationships with different health outcomes attributed to COVID-19 but could not present a strong and reliable model for the estimation.

Figure 6 was conducted to show the quantitative relationships (equations) and coefficients of determination ($R^2$) between NRC, ND cases and influencing parameters in Iran. The strong and reliable model ($R^2 \geq 0.7$) was conducted by air temperature in order to estimate number of NRC and ND in Ahvaz. The reason for the strong relationship of air temperature with NCC and ND is probably due to the fact that people in warm season in equatorial regions prefer to stay in closed environments while air conditioners work through a closed cycle and cause the spread of the virus to the whole environment.

However, Khuzestan province showed a higher number of AD with about 733 patients, North Khorasan province showed the highest rate of AD per its population so that the rate of $AD/10^5$ population was 30 in this province. This ratio calculated 21 accumulated deaths per $10^5$ population in Khuzestan (Table 5). The rate of $AD/10^5$ population was low and mainly ranged between 6 to 11 in other coastal and dry land regions. On the other hand, the AD ratio per $10^5$ population was high in mountainous provinces at around 15. In addition, the rate of new recovered cases per $10^5$ population was in the best situation in the most engaged province (Khuzestan) and Lorestan. The $NRC/10^5$ population ratio found about eight at that location. This ratio was four times lower in the rest of the country. The

worst situation in terms of NRC/$10^5$ population was found in Kerman province with less than 1 NRC/$10^5$ population in a day.

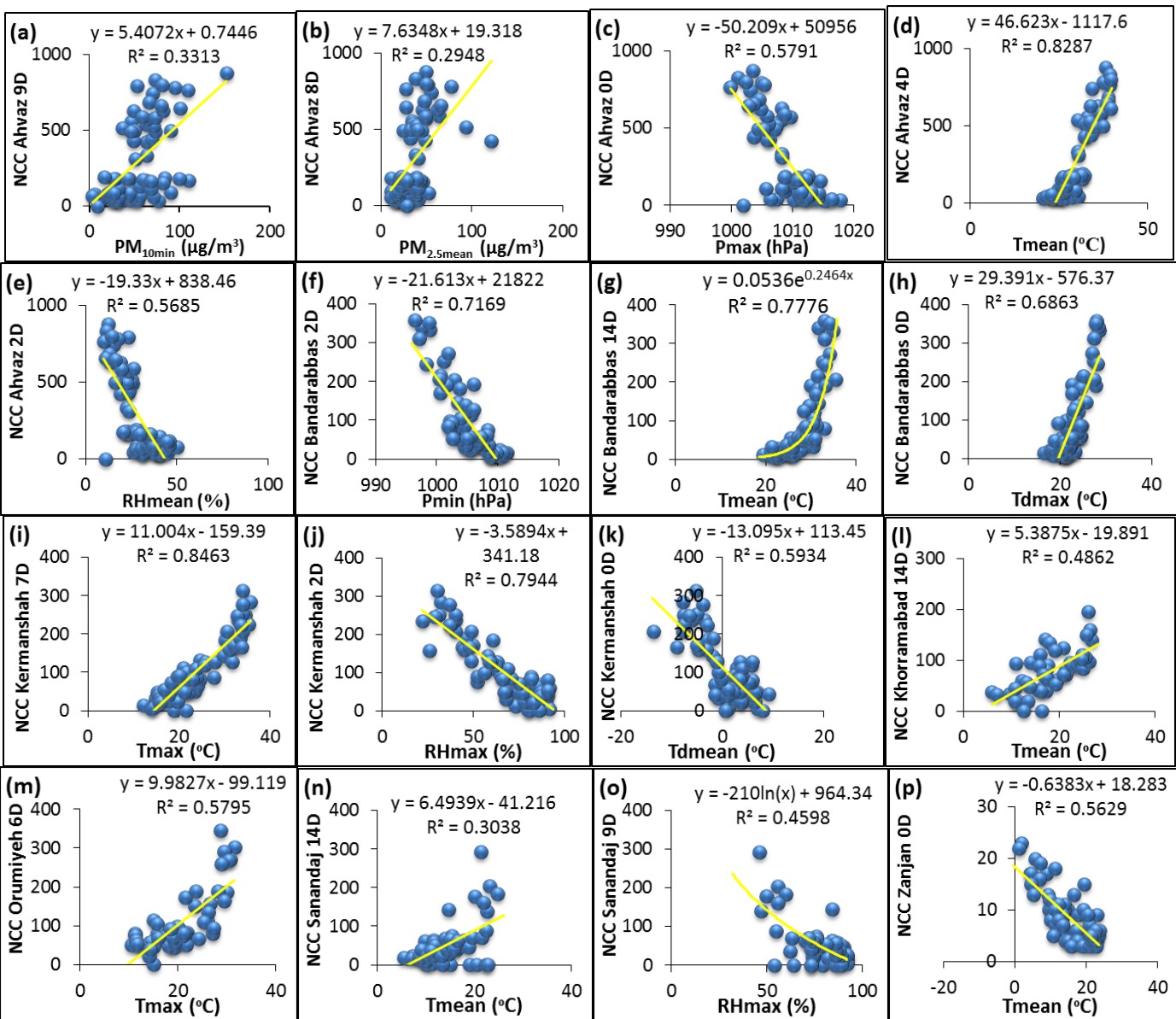

**Figure 5.** Best fit equation and $R^2$ from scatter plots of new confirmed cases (NCC) associated with COVID-19 versus air pressure, air temperature, relative humidity and dew point in (**a**–**e**) Ahvaz, (**f**–**h**) Bandrabbas, (**i**–**k**) Kermanshah, (**l**) Khorramabad, (**m**) Orumiyeh, (**n**,**o**) Sanandaj and (**p**) Zanjan during the study period.

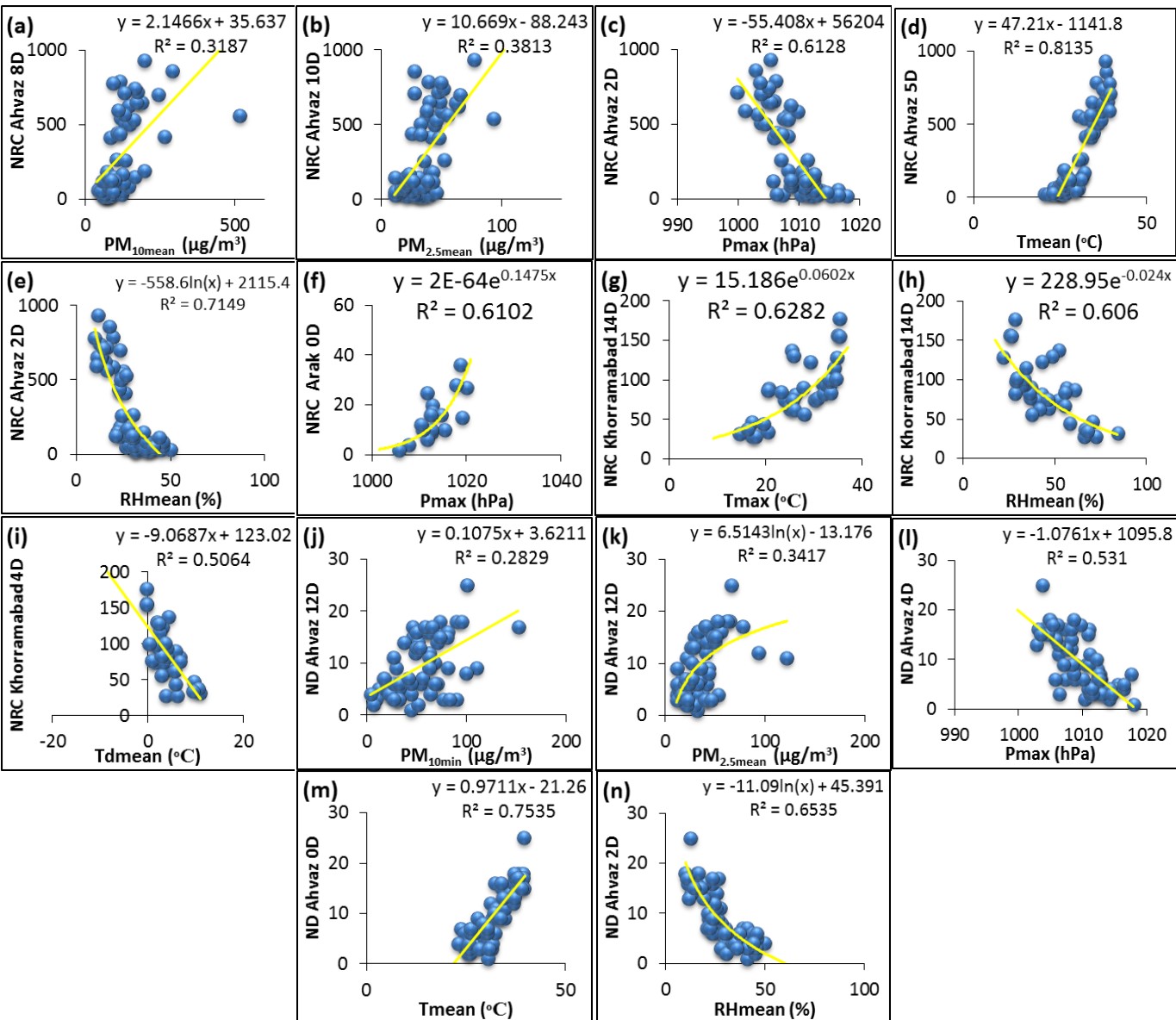

**Figure 6.** Best fit equation and $R^2$ from scatter plots of new recovered cases (NRC) in (**a–e**) Ahvaz, (**f**) Arak and (**g–i**) Khorramabad, and new deaths (ND) in (**j–n**) Ahvaz associated with COVID-19 versus air pressure, air temperature and relative humidity during the study period.

## 4. Conclusions

In this study, NCC, ACC, NRC, ARC, ND and AD were evaluated under the impact of geographical properties, meteorological parameters and air particulate matters in Iran. Provinces in dry lands (i.e., Kerman and South Khorasan) not only showed a low number of health endpoints associated with COVID-19 but also presented lower rates of health outcomes per $10^5$ population. Air temperature showed positive and significant correlation with the number of COVID-19 cases. This event is due to hot outdoor air, especially in coastal and equatorial regions, that forces people to stay at closed environment with no ventilation and closed-cycle air conditioner. Maximum air pressure was found to be the most frequent (66%) and significant parameter correlating with health outcomes associated with COVID-19. The following parameter was visibility with 54% frequency. Air pressure and relative humidity showed negative correlations with the spread of coronavirus. Wind speed and visibility with 1 m/s and 10,000 m are attributed to higher number of coronavirus infectious. The most engaged province in this study was Khuzestan with 22,077, 20,627

and 733 COVID-19 cases for ACC, ARC and AD, respectively. In addition, the highest rate of accumulated and new recovered cases per $10^5$ population were found in Khuzestan and Kerman provinces. North Khorasan also showed the worst rate of N&ARC/$10^5$ population. For future studies, it is necessary to evaluate COVID-19 cases though longer study periods as well as wider range of air pollutants in all provinces in Iran.

**Author Contributions:** Conceptualization, M.J.M., V.H. and H.M.; Data curation, H.M.; Funding acquisition, G.G.; Investigation, G.G.; Methodology, A.A.B.; Project administration, G.G.; Validation, M.J.M.; Visualization, H.M.; Writing—original draft, H.M. All authors have read and agreed to the published version of the manuscript.

**Funding:** This research was financially supported by NIMAD with the grant award number of 996638.

**Institutional Review Board Statement:** The ethical code of this study is *IR.NIMAD.REC.1399.224* from Iran National Committee for Ethics in Biomedical Research.

**Informed Consent Statement:** Not applicable.

**Data Availability Statement:** The data that support the findings of this study are available from the corresponding author upon reasonable request.

**Acknowledgments:** Research reported in this publication was supported by Elite Researcher Grant Committee under award number [996638] from the National Institute for Medical Research Development (NIMAD), Tehran, Iran.

**Conflicts of Interest:** The authors declare that they do not have any conflict of interest.

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
