# Peer review of "Geographical and Meteorological Evaluations of COVID-19 Spread in Iran"

_sustainability, doi:10.3390/su14095429_

Round 1

Reviewer 1 Report

  • In Abstract: Accumulated Confirmed Cases must be abbreviated ACC, not ARC;
  • Discussion is completely absent.

Author Response

Dear Reviewer #1,

Thank you very much for your comments and valuable suggestions about our manuscript. We appreciate your suggestions and tried to improve the revised draft. The following are our responses and corrections to your questions.

  • In Abstract: Accumulated Confirmed Cases must be abbreviated ACC, not ARC;

We thank the reviewer comment. It was edited accordingly.

  • Discussion is completely absent.

We improved discussion between Figure 3 and Table 2.

Reviewer 2 Report

This study aims to evaluate the possible relationship between visibility and meteorological parameters with different types of Covid-19 cases including new confirmed cases (NCC), accumulated confirmed cases (ACC), new recovered cases (NRC), accumulated recovered cases (ARC), new deaths (ND) and accumulated deaths (AD).

Before reading, I had high expectations from this work - unfortunately they did not come true. I have a number of critical comments - they are presented below.

Specific comments

1

line 42-44 -- Good physical condition is a considerable parameter for preventing of getting infected by the virus so that elder people and people with existing health conditions including 43 hypertension, chronic obstructive pulmonary disease, cerebrovascular disease and diabe- 44 tes are the most-at-risk portion of population [6,7].

- this is a bad statement and citation - authors should re-read the cited sources - - what is it talking about? - about the risks of infection or all the same about the risks of severe consequences for those already infected?

2

line 77 - This study aims to evaluate the possible relationship between visibility and meteorological parameters with different types of Covid-19 cases including …

From the whole introduction there is no indication of the importance in examining the visibility parameter.

All this needs to be substantiated in the text - probably the authors meant air pollution - - but ... and this needs to be substantiated too.

3

Line 80 - The current study conducted in 15 provinces during 3 months.

I do not understand the choice of the authors - - why three months? Moreover, it follows from the text that the interval April 3rd 2020 and June 11th 2020 was chosen - these are two months and 8 days. -- this is a very short period of time for study the epidemiological processes.

Why were these months chosen and not others?

4

Line 96-Figure 1.

the picture has a poor resolution (perhaps as a result of the PDF conversion ….)

5

line 139 -Figure 2.

The proportions of all the drawings are violated - the fonts are floating.

Image resolution is not good enough.

In my opinion, the data in the figures look strange / unexpected.

I would like to see - -the total number of tests done in these locations --

how many percent of the tests gave a positive result, etc. — so that these data can be evaluated and correlated with other known and confirmed data from other countries.

6

Line 145 -Figure 3.

The proportions of all the drawings are violated - the fonts are floating.

Image resolution is not good enough.

7

Line 150 — Amato et al 2022 also showed airborne transmission of Covid-19 through total suspended particles especially by micro plastics [26].

THIS is gross ignorance to cite the literature by this way. There is no evidence for this claim. No one has said this and it cannot be cited!

I studied this work - - this work is an explicit hypothesis. - it is of a pronounced local character (the area of one hospital) - does not at all reflect the possibility of transmitting a viable virus - I think it is impossible to cite this work in current study.

In fact, such citations are a big problem now - and this was clearly reflected in recent work -- .https://doi.org/10.1016/j.envres.2021.112116

8

Line 151-153 — In the mentioned study, negative and significant 151 relationship was found between relative humidity and air temperature with the prevalence of coronavirus.

@the prevalence of coronavirus.@  — What do the authors mean by this wording? - - the prevalence of the virus genome in the air - - or the spread of the disease?

although in the same paragraph it was already said about: "transmission of Covid-19 through total suspended particles".

I think the whole paragraph is very bad in terms of facts and understanding of the processes of the spread of diseases.

9

Figure 4.

The proportions of all the drawings are violated - the fonts are floating.

Image resolution is not good enough.

It is not clear from the figure which city belongs to which province.

10

Figure 5 and Figure 6.

I don't understand the logic of these figures - in particular, why is there no analysis for Tmin?

Conclusion

1 Poor literature review and poor introduction section

what is the logic to look for a correlation between all the parameters that are at the work studied - - just because others do it?

I think it's important - - briefly explain each option:

why air pollution? (how can it be related to covid)

why temperature? (how can it be related to covid)

why visibility? (how can it be related to covid)

etc

2 There are no graphs of air pollution levels for the provinces in the article (although this is the most important parameter) - this does not allow to visually assess the overall picture.

Author Response

Dear Reviewer #2,

Thank you very much for your comments and valuable suggestions about our manuscript. We appreciate your suggestions and tried to improve the revised draft. The following are our responses and corrections to your questions.

This study aims to evaluate the possible relationship between visibility and meteorological parameters with different types of Covid-19 cases including new confirmed cases (NCC), accumulated confirmed cases (ACC), new recovered cases (NRC), accumulated recovered cases (ARC), new deaths (ND) and accumulated deaths (AD).

Before reading, I had high expectations from this work - unfortunately they did not come true. I have a number of critical comments - they are presented below.

Specific comments

1

line 42-44 -- Good physical condition is a considerable parameter for preventing of getting infected by the virus so that elder people and people with existing health conditions including 43 hypertension, chronic obstructive pulmonary disease, cerebrovascular disease and diabe- 44 tes are the most-at-risk portion of population [6,7].

- this is a bad statement and citation - authors should re-read the cited sources - - what is it talking about? - about the risks of infection or all the same about the risks of severe consequences for those already infected?

In this sentence we stated that existing health problems (comorbidity) can increase the Covid-19 infectious rate. 

2

line 77 - This study aims to evaluate the possible relationship between visibility and meteorological parameters with different types of Covid-19 cases including …

From the whole introduction there is no indication of the importance in examining the visibility parameter.

All this needs to be substantiated in the text - probably the authors meant air pollution - - but ... and this needs to be substantiated too.

Dear reviewer, visibility has a close relationship with air particulate matters. Paragraphs 2 and 3 in introduction are about the effect of air pollution on coronavirus spread around the world. Since the access to air pollutants data are very limited in Iran. Visibility could play an alternative parameter for air particulate matters.

3

Line 80 - The current study conducted in 15 provinces during 3 months.

I do not understand the choice of the authors - - why three months? Moreover, it follows from the text that the interval April 3rd 2020 and June 11th 2020 was chosen - these are two months and 8 days. -- this is a very short period of time for study the epidemiological processes.

Why were these months chosen and not others?

We completely agree with the reviewer comments. Investigation the study period from late 2019 to the present time is the ideal study period for such studies. But in Iran there is not a clear procedure for receiving Covid-19 data. The investigated data in the present study are also collected from news websites like https://aftabnews.ir.

4

Line 96-Figure 1.

the picture has a poor resolution (perhaps as a result of the PDF conversion ….)

 Figure 1 has better resolution in Word file however in PDF format the words, numbers and signs seem good too.

5

line 139 -Figure 2.

The proportions of all the drawings are violated - the fonts are floating.

Image resolution is not good enough.

In my opinion, the data in the figures look strange / unexpected.

I would like to see - -the total number of tests done in these locations –

There is not access to the number of tests to researchers in Iran. But when it is said that one patient has positive test, it means the patient did not have any previous health problems and he or she infected only by SARS-CoV-2. PCR was the least test for identification of SARS-CoV-2 RNA.

how many percent of the tests gave a positive result, etc. — so that these data can be evaluated and correlated with other known and confirmed data from other countries.

 We enhanced the figure quality and hope it is satisfying.

6

Line 145 -Figure 3.

The proportions of all the drawings are violated - the fonts are floating.

Image resolution is not good enough.

 We enhanced the figure quality and hope it is satisfying.

7

Line 150 — Amato et al 2022 also showed airborne transmission of Covid-19 through total suspended particles especially by micro plastics [26].

THIS is gross ignorance to cite the literature by this way. There is no evidence for this claim. No one has said this and it cannot be cited!

I studied this work - - this work is an explicit hypothesis. - it is of a pronounced local character (the area of one hospital) - does not at all reflect the possibility of transmitting a viable virus - I think it is impossible to cite this work in current study.

In fact, such citations are a big problem now - and this was clearly reflected in recent work -- .https://doi.org/10.1016/j.envres.2021.112116

We thanks the reviewer comment. We rewrite the sentence.

8

Line 151-153 — In the mentioned study, negative and significant 151 relationship was found between relative humidity and air temperature with the prevalence of coronavirus.

@the prevalence of coronavirus.@  — What do the authors mean by this wording? - - the prevalence of the virus genome in the air - - or the spread of the disease?

It is the prevalence of the virus genome in the air. Because in Table 3 of the mentioned study the dependent variable is envelope protein (genomic units per TSP).

although in the same paragraph it was already said about: "transmission of Covid-19 through total suspended particles".

I think the whole paragraph is very bad in terms of facts and understanding of the processes of the spread of diseases.

9

Figure 4.

The proportions of all the drawings are violated - the fonts are floating.

Image resolution is not good enough.

It is not clear from the figure which city belongs to which province.

  We enhanced the figure quality and hope it is satisfying.

10

Figure 5 and Figure 6.

I don't understand the logic of these figures - in particular, why is there no analysis for Tmin?

It is true that Pearson correlation showed significant relationship between Covid-19 data and majority of meteorological parameters but in our opinion it is not reliable yet to say they can cause the increase or the decrease of Covid-19. Therefore, Figure 5 and Figure 6 seem necessary to show which parameters have stronger relationship (R2>0.7). As a result of these figures, we can see that air temperature, air pressure and relative humidity showed the strongest relationships. Surprisingly, these strong correlations happened in coastal regions. Moreover, from spring to autumn air conditioners work in all buildings through a closed cycle of indoor air in coastal regions. If somebody infected by coronavirus whom is in the room. The virus spreads so fast.  

We analyzed Tmin too. But in no cases the relationship between Tmin and Covid-19 data was higher than the relationship of Tmean and Tmax with Covid-19 data. Therefore, it was not consider in the manuscript.

Conclusion

1 Poor literature review and poor introduction section

what is the logic to look for a correlation between all the parameters that are at the work studied - - just because others do it?

I think it's important - - briefly explain each option:

why air pollution? (how can it be related to covid)

why visibility? (how can it be related to covid)

etc

Many studies showed that increase of air pollution could cause the higher spread of coronavirus. In this study significant correlation was found between PMs and covid-19 data in Ahvaz city but the correlations were not strong enough (R2<0.7). Therefore it can be say PM10 and PM2.5 could not directly and shortly caused the increase of Covid-19 data. For the rest of parameters except air temperature, air pressure and relative humidity it also revealed that they could not directly and shortly caused the increase of Covid-19 data.

why temperature? (how can it be related to covid)

Since air temperature, air pressure and relative humidity showed strong relationship (R2>0.7), we think it is worth asking how? The followings are our reasons:

From spring to autumn air conditioners work in all buildings through a closed cycle of indoor air in coastal regions. If somebody infected by coronavirus whom is in the room. The virus spreads so fast. Our hypothesis for air pressure is that It might increase the number of viruses per cubic meters of air. Relative humidity which is low in cities with dry climate like Kerman and Birjand could be the reason for low rate of coronavirus spread.

2 There are no graphs of air pollution levels for the provinces in the article (although this is the most important parameter) - this does not allow to visually assess the overall picture.

The levels of fine and coarse air particulate matters (PM2.5 and PM10) are shown in Figure 5 and Figure 6 in Ahvaz city. Since they did not show strong correlation with Covid-19 data like air temperature, air pressure and relative humidity, we did not consider their temporal variations in the study.

Submission Date

12 March 2022

Date of this review

22 Mar 2022 12:23:48

Reviewer 3 Report

In the study, authors revealed the impact of geographical properties, meteorological parameters and air particulate matters in Iran. Interesting and important results in terms of data are presented. However, there are many factors that increase or decrease the spread of the COVID-19 virus. It is not correct to attribute this situation only to geographical conditions. For example, was the vaccine administered at the time of the study? How much social quarantine has been implemented during this period? How many days was it made? Are schools kept open? Have there been any changes in practice in the measures in public places? Therefore, presenting the study data as if they are related creates scientific bias. If the study authors present the study descriptively, they will present much more valuable data from a scientific point of view. In this state, they will be trying to prove a very assertive and compelling result. The discussion section of the study is insufficient. For this reason, revising the study and presenting it in another format will provide the value it deserves. Best regards

Author Response

Dear Reviewer #3,

Thank you very much for your comments and valuable suggestions about our manuscript. We appreciate your suggestions and tried to improve the revised draft. The following are our responses and corrections to your questions.

In the study, authors revealed the impact of geographical properties, meteorological parameters and air particulate matters in Iran. Interesting and important results in terms of data are presented. However, there are many factors that increase or decrease the spread of the COVID-19 virus. It is not correct to attribute this situation only to geographical conditions. For example, was the vaccine administered at the time of the study? How much social quarantine has been implemented during this period? How many days was it made? Are schools kept open? Have there been any changes in practice in the measures in public places? Therefore, presenting the study data as if they are related creates scientific bias. If the study authors present the study descriptively, they will present much more valuable data from a scientific point of view. In this state, they will be trying to prove a very assertive and compelling result. The discussion section of the study is insufficient. For this reason, revising the study and presenting it in another format will provide the value it deserves. Best regards

We agree with the reviewer comment that visibility, air particulate matters and majority of meteorological parameters could not be a reliable parameter (R2<0.7) for direct estimation of Covid-19 data although Pearson correlation showed significant relationship. Therefore, one reason for Figure 5 and Figure 6 is to distinguish the unsuitable parameters. But it is logical to ask why air temperature, air pressure and relative humidity showed strong relationship (R2>0.7) in some cities (mainly in coastal cities). In coastal regions air conditioners work in all buildings through a closed cycle of indoor air from spring to autumn in Iran. If somebody infected by coronavirus whom is in the room. The virus can spread so fast. Our hypothesis for air pressure is that It might increase the number of viruses per cubic meters of air. Relative humidity which is low in cities with dry climate like Kerman and Birjand could be the reason for low rate of coronavirus spread.

We also improved the discussion part between Figure 3 and table 2 according to your constructive comments.

Submission Date

12 March 2022

Date of this review

22 Mar 2022 13:15:06

Round 2

Reviewer 1 Report

/

Author Response

Dear Reviewer #1,

We thank and appreciate the Reviewer’s valuable time and constructive comments for reviewing our manuscript.   

Reviewer 2 Report

I think the work can be accepted for publication after minor revision. 

line 42-44 -- the sources cited by the authors do not mention a high risk of infection - it says about the high risk of severe disease and mortality among these patients/ 

a high risk of infection depends on the frequency of contacts between people and social activity - 

being infected and having a severe course of the disease are different concepts.

References

many links to the source do not work - you need to check if there was a failure when converting the file.

Author Response

Dear Reviewer #2,

Thank you very much for your comments and valuable suggestions about our manuscript. We appreciate your suggestions and tried to improve the revised draft. The following are our responses and corrections to your questions. 

line 42-44 -- the sources cited by the authors do not mention a high risk of infection - it says about the high risk of severe disease and mortality among these patients/ 

a high risk of infection depends on the frequency of contacts between people and social activity - 

being infected and having a severe course of the disease are different concepts.

 Dear Reviewer we checked the manuscript. The words “high risk” or “high risk of infection” did not mentioned in the manuscript.

References

many links to the source do not work - you need to check if there was a failure when converting the file.

Thank you very much for the constructive comment, we corrected them accordingly. The style of all references also changed to Vancouver. 

Reviewer 3 Report

Dear Authors,

I still stand by my previous assessment. There are many factors in the spread of the coronavirus, and it would not be scientifically correct to associate it with only weather-related parameters. Yes, this can be an influencing factor, but it is not the only factor. If the aim and design of the study is in the form of presentation of the parameters related to the weather, it will make more scientific contribution in the next process. Best regards

Author Response

Dear Reviewer #3,

Thank you very much for your comments and valuable suggestions about our manuscript. We appreciate your suggestions and tried to improve the revised draft. The following are our responses and corrections to your questions.

Dear Authors,

I still stand by my previous assessment. There are many factors in the spread of the coronavirus, and it would not be scientifically correct to associate it with only weather-related parameters. Yes, this can be an influencing factor, but it is not the only factor. If the aim and design of the study is in the form of presentation of the parameters related to the weather, it will make more scientific contribution in the next process. Best regards

Dear reviewer, the aim of the current study is to evaluate the association of meteorological parameters and geography with Covid-19 in Iran. We would be thankful for receiving more constructive comments.
